# Use of Microbial Consortia in Bioremediation of Metalloid Polluted Environments

**DOI:** 10.3390/microorganisms11040891

**Published:** 2023-03-30

**Authors:** Elham Lashani, Mohammad Ali Amoozegar, Raymond J. Turner, Hamid Moghimi

**Affiliations:** 1Extremophiles Laboratory, Department of Microbiology, School of Biology and Center of Excellence in Phylogeny of Living Organisms, College of Science, University of Tehran, Tehran 14178-64411, Iran; elham.lashani92@ut.ac.ir; 2Microbial Biochemistry Laboratory, Department of Biological Sciences, University of Calgary, 2500 University Dr. NW, Calgary, AB T2N 1N4, Canada; turnerr@ucalgary.ca; 3Department of Microbiology, School of Biology, College of Science, University of Tehran, Tehran 14178-64411, Iran

**Keywords:** metalloids, microbial bioremediation, microbial consortia, microbial ecology, tolerance gene and protein

## Abstract

Metalloids are released into the environment due to the erosion of the rocks or anthropogenic activities, causing problems for human health in different world regions. Meanwhile, microorganisms with different mechanisms to tolerate and detoxify metalloid contaminants have an essential role in reducing risks. In this review, we first define metalloids and bioremediation methods and examine the ecology and biodiversity of microorganisms in areas contaminated with these metalloids. Then we studied the genes and proteins involved in the tolerance, transport, uptake, and reduction of these metalloids. Most of these studies focused on a single metalloid and co-contamination of multiple pollutants were poorly discussed in the literature. Furthermore, microbial communication within consortia was rarely explored. Finally, we summarized the microbial relationships between microorganisms in consortia and biofilms to remove one or more contaminants. Therefore, this review article contains valuable information about microbial consortia and their mechanisms in the bioremediation of metalloids.

## 1. Introduction

### 1.1. Metalloids

Metalloid elements have some physical and chemical characteristics between metal and nonmetal elements. Metalloids included arsenic (As), antimony (Sb), boron (B), germanium (Ge), silicon (Si), tellurium (Te), and polonium (Po)—although we think less of this element in biology due to its radioactivity. Astatine (At) and selenium (Se) are also sometimes considered as they have metalloid characteristics under certain conditions. Metalloids are exploited in medicine [1,2,3,4], semiconductor devices [5,6,7], glasses and ceramics [8,9], solar cells/batteries [10,11], specific polymers [12], construction material, and agricultural applications [11,13]. B, Si, and Se have essential functions in most organisms, including humans, as essential trace elements. However, high levels of these elements are toxic. While As, Sb, Te, and Ge are more toxic [14].

### 1.2. Prevalence and Toxicity of Metalloids

#### 1.2.1. Selenium

Selenium was discovered by Jons Jacob Berzelius in 1808 and derived from the Greek word selene which means the moon. This element is the 69th most abundant element on the earth and belongs to the group V periodic table [15]. It exists in different oxidation states such as selenite (IV; SeO_3_^2−^), selenate (VI; SeO_4_^2−^), elemental Se (0), and organic forms such as dimethyl selenide, methyl selenide, selenomethionine, selenocysteine. According to Paul and Saha (2019), California (38%), Ireland (32%), Punjab (8%), Jaipur (9%), and China (2%) are the most selenium-polluted regions in the world. Selenium is released to the environment by natural and anthropogenic activities such as volcanic eruption, mining, weathering of rocks, coal mining (so-called selenium curse of the eastern range of the north American Rocky mountains) and combustion, and effluent waste by some industries and agriculture leading to polluted areas [16,17,18,19,20,21,22].

There is a narrow limit between selenium toxicity and the amount required for human health. Paradoxically, less than 40 µg/day of selenium is essential for the body and more than 400 µg/day is toxic to the human body [23]. Daily intake of selenium in food varies from 0.055 to 0.4 mg per day, required for crucial body functions such as antioxidant defense, protein folding, and cell signaling [24,25]. Some primary selenoprotein genes in mammals have central roles in Redox signaling (GPX1, GPX3, GPX4, TRXRD1, TRXRD2), Protein folding and degradation (SEP15, SELS), and metabolism (SEP1, SPS1, SPGS) [26,27]. Selenium deficiency is associated with Keshan disease [28,28,29,30,31], muscle weakness [32], Kashin Beck [33,34,35,36,37], cardiomyopathy [38,39,40,41], and redox dysregulation [42]. In contrast, exposure to excessive amounts of selenium can lead to disorders such as selenosis, loss of hair and nails, redox dysregulation, mitochondrial dysfunction, and cell growth inhibition [43,44,45,46,47,48].

#### 1.2.2. Arsenic

Arsenic (As, group V periodic table) is the 20th most abundant elements in the earth’s crust with a terrestrial abundance of about 1.5–3 mg/kg and an average abundance of about 5 mg/kg [15]. Anthropogenic and natural activities are the main sources of arsenic pollution in the world. The groundwater of different regions of Asia (Bangladesh [49,50,51,52], India [53,54,55,56,57,58], China [59,60,61,62,63,64], Nepal [65,66], Cambodia [67], Vietnam [68], Myanmar [69], Pakistan [70,71,72,73,74,75], and Indonesia [76]), North and South America (USA [77,78], Canada [79,80], Argentina [81], Chile [82], and Mexico [83]), Europe (Hungary [84]) and Africa (South Africa [85]) are contaminated by arsenic [86]. Arsenic occurs in different oxidation states in nature, including arsenate (+5, AsO_4_^3−^), arsenite (+3, AsO_3_^3−^), elemental arsenic (0), and arsenide (−3). Akin to SeO_3_^2−^, AsO_3_^3−^ is the most toxic form among arsenic oxyanions in the environment [87,88]. Arsenic-containing compounds were applied in the manufacture of glass [89], semiconductors [90] and alloys [91], herbicides [92,93], wood preservatives [94], pesticides [95], animal feed additives [96], and medicine [97]. Disruption in cell signaling [98], reactive oxygen species (ROS) generation [99], high affinity to protein thiols or vicinal sulfhydryl groups [100], interruption in the binding of some hormones to their receptor [101], and prevention of oxidative phosphorylation [102] are the main effects of arsenic oxyanion on cells [88].

Depending on the concentration, oxidation state, and exposure time, arsenic can cause health problems such as cancer (skin [103], lung [104,105], bladder [106], and liver [107]), skin lesions [108], PNS (Peripheral Nervous System) disorder [109], liver failure [110], leukopenia [111], circulatory disease [112,113], anemia [114,115], and death [116,117].

#### 1.2.3. Boron

Boron, the fifth element in the periodic table (group III), is a ubiquitous element in the environment that comprised an average concentration of around 10 mg/kg of the earth’s crust. Na_2_B_4_O_5_(OH)_4_·8H_2_O is a common form of boron in ores widely distributed in California and Turkey [118]. Argentina, Russia, Chile, Peru, China, Libya, Egypt, Iraq, Morocco, and Syria are other countries containing many boron deposits [119,120]. This element forms approximately 230 compounds, and natural and anthropogenic activities such as volcanoes, commercial uses, fertilizers, wastewater treatment plants, forest fires, and coal combustion can release it into the atmosphere [121]. Boron has very useful applications in some industries such as the manufacture of glass and ceramics, fertilizer and detergent [122,123,124,125].

This essential element plays a pivotal role in immune response [126,127], mineral metabolisms [128], and the endocrine system [129]. It can also inhibit osteoporosis in postmenopausal women [130] and decrease cardiovascular disease [131]. Similar to other metalloids, it can be toxic in higher concentrations and causes many problems such as an increase in the oxidative state of a cell, DNA damage, impairment of DNA repair systems and membrane functions, or the inhibition of protein folding, protein function, and activities in living organisms [119,132,133]. Boron is found in some antibiotics such as boromycin [134], tartrolons [135], aplasmomycin [136], and biomolecules such as the bacterial autoinducer 2 (AI-2] vibrioferrin (a siderophore) [137] and borolithochromes (pigment in algae *Solenopora jurassica*) [138]. Additionally, some nitrogen-fixing bacteria need boron as a cofactor for growth and nitrogen fixation [119].

#### 1.2.4. Antimony

Antimony belongs to subgroup 15 of the element periodic table (atomic number 51) with average concentrations less than 1 μg/L in nature. Most of the world’s antimony reserves are located in South Africa, China, Russia, Bolivia, Tajikistan, and Mexico [139]. Due to the chalcophilic nature of antimony and its presence in ores containing chalcogen, smelting, and mining of ores containing these compounds, especially sulfide ores [140] are among the polluting sources of this element [141].

Antimonate (Sb (V); Sb(OH)_6_^−^) and antimonite (Sb (III); Sb(OH)_3_) are the two common inorganic forms of antimony present in natural waters, and Sb(OH)_3_ is more toxic than the other one [142]. Antimony’s other toxic compound is antimony trichloride (SbCl_3_), used in alloys, as a constituent of paint pigments, and in rubber compounding. Other major applications of antimony are included in various industries and use such as semiconductor, alloys, batteries, catalyst, and medicine [143,144,145]. In ancient times, antimony was used to purify precious metals such as gold and silver. Furthermore, antimony is used with other compounds to make textiles, paper, and plastics as a fire retardant agent [140].

Antimony is not present in living systems and, such as arsenic, is highly toxic to humans and living organisms. Eye, skin, lung, mucous membrane irritation, oxidative DNA damage, pneumoconiosis, and increased lung, heart, and gastrointestinal diseases are some problems caused by long-term exposure to antimony [146,147]. Antimony can affect the nitrogen cycle in soil by influencing urease function under pH 7 [148].

#### 1.2.5. Tellurium

Tellurium oxyanions are another highly toxic metalloid that studies in this review. Tellurium is an element that belongs to the 16th group of the periodic table with atomic number 52 and has two allotropic forms, including white crystalline metal and black amorphous powder [149,150]. The concentration of tellurium in the earth’s crust is very low and about 1–5 µg/kg [151]. Tellurium is found as oxyanions tellurite (IV; TeO_3_^2−^) and tellurate (VI; TeO_4_^2−^). Tellurium is found in industries such as petroleum refining plants, glass, electronic and photoelectronic industries, optics, and sensors [152]. Tellurium can be found in a variety of ores as well as coal. Another tellurium application is in medicine and has traditionally been exploited as an antimicrobial agent in treating some infectious diseases, including leprosy, tuberculosis, dermatitis, cystitis, and severe eye infections [153]. Tellurium is also used in labeling, imaging, and targeted drug delivery systems and has some anti-inflammatory, anti-fungal, anti-leishmaniasis, and immunomodulatory activities [154,155,156,157,158,159]. Exposure to a high amount of tellurium can cause several health issues, such as respiratory irritation, headache, drowsiness, weakness, malaise, lassitude, gastrointestinal symptoms, dizziness, and dermatitis [160]. Both Se and Te are mixed with Cadmium to make quantum dots (used in phone and TV screens) and photoreceptors in solar cells leading to concerns of their disposal and subsequent release into water systems [161].

#### 1.2.6. Other Metalloids

Germanium is another metalloid that belongs to group 14 and period 4 of the periodic element table. This metalloid is ranked 54th among the most abundant elements in the earth’s crust and has two stable oxidation states +2 and +4 in nature. Only a few compounds of germanium such as GeO_2_, GeH_4_, GeCl_4_, and GeF_4_ have toxic properties and their organic forms have no effect on human health. Due to similar outer electron structure and properties, Germanium is also called a pseudo isotope of silicon [162,163]. Germanium is used in small quantities in some fields such as fiber optics [164,165], micro- and nano electronics [166], infrared detectors [167], and polymerization catalysts [168].

After oxygen, silicon (Si) with 27% is the most abundant element in the earth’s crust. This element is found in many human organs and its deficiency is related to infection and bone weakness. Si is mostly biologically inert, and it can be used as a drug carrier in ointments and hydrogel coatings in medical devices [169]. There are no specific reports on the devastating effects of astatine and silicon in the literature.

### 1.3. Methods of Bioremediation

Several methods have been suggested to eradicate metalloid contamination in various environments and locations including physical, chemical, and biological methods. All methods remediate and detoxify; however, bioremediation is beneficial for removing the toxic effects of metalloids and recovering them for industrial applications. This method gains increasing interest in recent years due to its cost-effectiveness, eco-friendly, and simple operable manner. Furthermore, this method needs no special technical equipment/skills and typically does not result in any additional waste and toxic compounds. Bacteria have the primary role in microbial bioremediation because of their small size, high numbers, fast growth, suitable adaptation to different environments, and manageable operating and culture condition. The main factors affecting bioremediation are included, organometallic ions, soil type, type and concentration of contaminants, nutrient availability, redox potentials (Eh), pH, moisture, solubility, microbial community and interaction, electron acceptors, and temperature [170,171,172].

Bioremediation is categorized into in-situ and ex-situ methods (for further information on this process please see [171,173]). Using fungi in a process called mycoremediation, is another approach for the effective eradication of metalloids. Bioremediation with algae and microalgae is called phytoremediation. Phytoremediation refers to bioremediation based on plants, a category that utilizes pollutant hyperaccumulator plants. In combination with plants (phytobial remediation), endophytic bacteria include another biological approach used for the bioremediation of polluted regions [174,175,176,177].

The main aim of bioremediation is to decrease the bioavailability of contaminants in environments and minimize their toxic and adverse effects by using living organisms, specifically bacteria and fungi [178,179]. There are different mechanisms for removing metal/metalloids from contaminated areas, such as bioaccumulation, biosorption, biotransformation, bioleaching, biomineralization, and participating microorganisms in metal/metalloid chemisorption. Biosorption is an energy-independent and reversible process in which pollutant ions interact with specific biochemical functional groups exposed on the surface of the cell wall and outer membrane of microorganisms [180]. Bioaccumulation is an energy-dependent process, which can be irreversible, that ions enter into the cell and accumulate intracellularly after binding to the cell via some particular transferring proteins and channels [181]. Nowadays, interest focuses on biotransformation, a process based on oxidation-reduction reactions. Since the toxicity of metalloids depends on their oxidation states and can be changed by losing or adding electrons, metalloid-reducing and oxidizing microorganisms gain much attention in the recent investigation on bioremediation of contaminants [182]. The main mechanisms of metal resistance in microorganisms are summarized in Figure 1 [183].

Bioleaching refers to a process using microorganisms to alter the solubility of ions and make them soluble forms to facilitate their removal from an ore [184]. Another process based on precipitation called biomineralization can be applied for increasing the effectiveness of other mechanisms for eradicating metal/metalloid toxicants [185]. The latter mechanism refers to facilitating and accelerating the rate of chemisorption and precipitation with microorganisms. A review article by Shahid Sher and Abdul Rehman (2019) described these bioremediation strategies for arsenic [177].

Here in this review, we examine the biodiversity and ecology of bacteria and fungi involved in metalloid bioremediation. A few similar articles are studying both the biodiversity of microorganisms present in metalloid-contaminated areas and the genes, transporters, and mechanisms involved in metalloid bioremediation. Most of these studies usually investigated only one metalloid, such as arsenic or selenium. Furthermore, microbial communication in consortia and microbial aggregates is an important issue that plays a crucial role in the superiority of a consortium over culturing one microorganism for removing metalloids, which received little attention in papers. Therefore, genes and proteins involved in the various bioremediation processes using bacteria and fungi are overviewed. Finally, the review summaries the progress in metalloids’ bioremediation and mechanisms participating in this process by microbial aggregates, biofilm, and consortia.

## 2. Ecology and Diversity of Metalloid-Resistant Microorganisms

### 2.1. Microbial Diversity of Metalloid-Contaminated Regions

Microorganisms adapt to the microenvironment, live together in consortia and biofilms, and interact with each other and abiotic parts of the ecosystems [186]. By studying microbial diversity and the basic scientific report, we can better understand microbial community structure and its function. Some anthropogenic activities such as mining, agriculture, and industrial wastewater, the release of organic and inorganic pollutants, pesticides, urban and rural affluent, city development, and so on, can affect microbial diversity and shift the community structure into specific bacterial and fungal taxa. Microbial inhabitants of these areas can tolerate high levels of pollutants, and some of them have a crucial role in nutrient cycling and remediation of contaminated regions [187]. Below follows examples of microbial diversity in relationship to oxyanion pollutants and bioremediation.

#### 2.1.1. Microbial Communities in Selenium Oxyanion-Contaminated Sites

Due to explosive activities in mines, Se oxyanions can be associated with nitrate (NO_3_^−^). Subedi et al. (2017) used a bacterial consortium from two mine sites in Canada to investigate simultaneous denitrification and SeO_4_^2−^ reduction. Metagenomics analysis of inoculum showed that *Pseudomonas*, *Lysinibacillus*, and *Thauera* were dominant operational taxonomic units (OTU) in biomass, respectively. The minor OTUs, such as *Exiguobacterium*, *Tissierella*, and *Clostridium*, were increased after the addition of NO_3_^−^ in culture media. Results indicated that a consortium including SeO_4_^2−^ and NO_3_^−^-reducing bacteria could be effectively applied for removing both SeO_4_^2−^ and NO_3_^−^ pollution at the same time [188].

Granular sludge from brewery wastewater containing microbial aggregates can be used to transform selenium oxyanion into selenium nanoparticles (SeNPs). The granular sludge’s microbial structure has been revealed by the high-throughput method, and the results showed that members of the families *Veillonellaceae* and *Pseudomonadaceae*, and *Clostridiaceae* are frequent in these aggregates. Both *Veillonellaceae* and *Clostridiaceae* can use lactate as a carbon source and electron donor and transform it into acetate, while *Pseudomonadaceae* members consume this acetate in a syntrophic relationship. Another interesting issue is the significant increase in the abundance of *Pseudomonas* species (from 0.03% of inoculum to 10% in culture medium) after three weeks of incubation in the presence of SeO_3_^2−^. Although the genus *Pseudomonas* is known primarily as an aerobic bacterium, its presence and increase in granular sludge as an anaerobic condition is a curious outcome. This is an example of metabolic cooperation between species present in a consortium. Outer membrane porin and cytoplasmic elongation factor Tu were associated with the selenium nanosphere, indicating intracellular nanoparticle biosynthesis and subsequent secretion [189].

In another study, SeO_3_^2−^-reducing bacteria were isolated from wastewater sludge in the presence of 100 mM SeO_3_^2−^ and lactate as an electron donor. *Citrobacter* sp. and *Providencia* were identified by 16S rRNA sequencing after spreading on BSM culture (basal salt medium) plates. Further studies on isolates revealed that acetate and lactate were better electron donors and carbon sources than propionate, butyrate, and glucose [190].

Thirty-eight selenium-resistant bacteria were isolated from different SeO_3_^2−^ and SeO_4_^2−^ polluted areas in India, ranging from 2.25–3 mg/kg in soil and 0.08–1.10 mg/L in water samples. Eight morphologically different strains were chosen for polyphasic taxonomy analysis, including biochemical tests, 16S rRNA sequencing, and FAME (Fatty acid methyl ester) methods. Two and three strains phylogenetically belong to the genera *Delftia*, and *Bacillus*, respectively. Other strains were related to the genera *Achromobacter*, *Pseudomonas*, and *Enterobacter*. Minimum inhibitory concentrations (MIC) of these strains were determined by 300–750 mM which showed the enormous potential of these microorganisms in bioremediation of selenium oxyanion polluted sites [191].

*Pseudomonas* is one of the most predominant genera present in the selenium oxyanion-contaminated regions. This bacterium can reduce SeO_3_^2−^ to SeNPs and is well known for its potential ability for a high amount of selenium oxyanion resistance. Because of its specific reductase enzymes and capability in NADPH equivalent production, *Pseudomonas* is a suitable candidate for the bioremediation of metalloids [192,193]. Another group is *Rhodococcus*, which has particular capabilities in the biotransformation of metalloids such as changes in the sensitivity of cellular targets, tolerance of high concentrations of metalloids, biodegradation of hydrocarbons, and the production of siderophores to bind to metalloids such as AsO_3_^3−^. In addition, *Rhodococcus* contains operon and genetic elements, mycothiol, phosphate transporters, which play an essential role in reducing the toxicity and biotransformation of metalloids, reducing oxidative stress, and the entry of metalloids into the cell, respectively. Other notable abilities of this bacterium include the bioconversion of the oxyanions into Se and Tellurium nanoparticles (TeNPs) and nanorods [25,149,194,195,196,197,198].

Sediment samples were collected from New Jersey (USA) and Chennai (India) and were enriched with 10 mM sodium SeO_4_^2−^ in MSM medium and pyruvate or 4-hydroxy benzoate as a carbon source. SeO_4_^2−^ was the sole electron acceptor in culture media. After culturing on solid media, isolated strains were selected considering different sizes, shapes, and morphology. A 16S rRNA analysis was applied for the molecular identification of selected strains. *Pseudomonas stutzeri* and three new species were characterized that phylogenetically related to *Geovibrio ferrireducens*, *Pelobacter acidigallici*, and *Chrysiogenes arsenatis* with 99, 98, 97, and 94% similarity, respectively. All strains prefer pyruvate as an electron donor in comparison with 3-hydroxy benzoate, benzoate, and 4-hydroxybenzoate. NO_3_^−^ and SeO_4_^2−^ also are electron acceptors in the respiration process. This study indicated that SeO_4_^2−^-respiring bacteria are phylogenetically and physiologically diverse and ubiquitous in aquatic sediments [199]. In another study conducted to evaluate selenium oxyanion pollutants, two phosphate mines in Idaho with different Se concentrations targeted high throughput sequencing to understand these reclaimed sites’ bacterial and fungal diversity. *Ascomycota*, *Basidiomycota*, *Zygomycota*, *Chytridomycota*, *Glomeromycota*, and *Rozellomycota* were abundant fungal phyla isolated from different Se-contaminated sites, respectively. In contrast, *Actinomycetota*, *Alphaproteobacteria*, *Bacteroidota*, *Acidobacteriota*, *Chloroflexota*, *Verrucomicrobiota*, *Planctomycetota*, *Deltaproteobacteria*, *Gemmatimonadota*, *Betaproteobacteria*, *Firmicutes*, and *Gammaproteobacteria* were dominant in bacterial strains. Microorganisms involved in the nitrogen cycle are dominant in sequences that indicate that there could be a connection between nitrogen and selenium cycles in this ecosystem [200].

#### 2.1.2. Bacterial Communities in Arsenic Sites

The contaminant concentration is a crucial factor that affects the microbial community structure of the polluted area [173]. In this regard, Valverde et al. (2011) studied the microbial community shift in response to different concentrations of arsenic in the Terrubias mine in Spain by the culture-dependent method. The results showed that *Bacillota* and *Gammaproteobacteria* are the most frequent taxa, and *Bacillus* and *Pseudomonas* are the dominant genera in this region, respectively. In the presence of a high concentration of arsenic in soil samples, the bacterial diversity of samples shifts to the preponderance of Firmicutes [201].

The microbial diversity of the As-contaminated groundwater of Assam (India) was studied by the culture-dependent method. In this study, 89 bacterial strains were isolated, and 16S rRNA sequencing analysis was applied for microbial community characterization. The results were shown in Table 1. *Brevundimonas* (35%) and *Acidovorax* (23%), *Acinetobacter* (10%), *Pseudomonas* (9%) were the dominant genera isolated from this region. The other genera include *Undibacterium*, *Herbaspirillum*, *Rhodococcus*, *Staphylococcus*, *Bosea*, *Bacillus*, *Ralstonia*, *Caulobacter*, and the member of the order *Hyphomicrobiales* [202].

Due to wastewater release in marine environments, the estuary is prone to contamination, including metalloids. In this regard, Jackson et al. (2005) studied AsO_4_^3−^-resistant bacteria isolated from Lake Pontchartrain estuary, Louisiana, USA. The amount of arsenic in samples was 7.02 µg/g sediment. Thirty-seven isolated strains were selected for molecular characterization. These isolates belong to the phyla or classes *Gammaproteobacteria*, *Betaproteobacteria*, *Alphaproteobacteria*, *Bacillota*, *Actinomycetota*, *and Bacteroidota*. The results are shown in Table 1 [203].

In the case of bioremediation, there are many studies investigating arsenic contamination in aquifer samples. Three groundwater samples from As-rich sites of Chianan plain, Taiwan were exploited for the detection diversity of dissimilatory AsO_4_^3−^ reducing microorganisms with the pyrosequencing method. In this study, *Bacillota* were dominant at Arsenic high concentration areas, and *Proteobacteria* was frequently observed in other groundwater samples with lower arsenic concentrations. The genera *Acinetobacter* and *Bacillus* are dominant in all three samples. The results showed that lactate is the favorable carbon and electron source and arsenic reducing bacteria constitute the leading performers, and sulfate-reducing bacteria (SRB) play a negligible role in the bioremediation of arsenic [204].

Molecular identification of arsenic-rich sites of the Latium region of Italy showed that *Gammaproteobacteria* are the dominant taxon in arsenic-tolerant microorganisms in thermal waters, while Alpha- and *Beta- proteobacteria* are prevailed in surface and ground waters [206]. In another study of arsenic-rich areas in Italy, the abundance of bacterial phyla and classes was as follows: *Gammaproteobacteria > Betaproteobacteria > Bacillota > Alphaproteobacteria > Bacteroidota. Pseudomonas and Comamonas* were the dominant genera in sequenced strains [206]. Most taxa in deep-sea sediments of the Indian Ocean Ridge were mainly related to *Proteobacteria* and *Actinomycetota*, especially *Microbacterium esteraromaticum* [207].

According to Table 1, different classes of *Proteobacteria* are dominant in arsenic-contaminated regions. Among them, *Alphaproteobacteria* and *Betaproteobacteria* are predominant in all samples. *Acinetobacter*, *Undibacterium*, *Brevundimonas*, *Janthinobacterium*, and *Bacillus* were the remarkable genera in all samples.

#### 2.1.3. Communities of Antimonite Sites

Samples were taken from Sb-contaminated areas of Guizhou province (China) to determine microbial diversity. The results indicated that the phyla *Pseudomonadota* (60.5% of all sequences) and *Actinomycetota* (10.3% of all sequences) were prevalent in all samples. *Chloroflexota*, *Cyanobacteria*, *Acidobacteriota*, and *Gemmatimonadota* were the other phyla presented in the samples. The most dominant taxa included *Thiobacillus*, *Limnobacter*, *Nocardioides*, *Lysobacter*, *Phormidium*, and *Kaistobacter*. Besides, archaeal phyla such as *Euryarchaeota* and *Crenarchaeota* comprised only 0.4% of all sequences [208]. In another study, Wang et al. (2018) investigated the microbial diversity of the Xikuangshang Sb mine in China which is known as the capital of antimony around the world. *Proteobacteria*, *Acidobacteriota*, *Chloroflexota*, *Bacteroidota*, *Actinomycetota*, *Gemmatimonadota*, and *Cyanobacteria* were abundant phyla detected from this region [209].

### 2.2. Metalloid-Resistant Bacteria from Metalloid-Contaminated Sites

Here we will overview specific species and strains isolated from metalloid-polluted sites that display strong resistance to this class of elements. Maltman et al. (2016) investigated the biodiversity of metalloid respiring bacteria associated with *Riftia pachyptila* inhabitants in hydrothermal vents. They isolated 107 bacteria, and 106 of them can use at least one metalloid oxyanion, such as TeO_4_^2−^, TeO_3_^2−^, SeO_4_^2−^, and SeO_3_^2−^, as a last electron acceptor. Epibiont diversity of different locations of Juan de Fuca Ridge Black Smoker Field showed the predominance of *Curvibacter*, *Shewanella*, *Pseudomonas*, and *Pseudoalteromonas* in Explorer Ridge and *Vibrio*, *Pseudoalteromonas*, *Curvibacter*, and *Shewanella* in Axial Volcano, respectively [210].

## 3. Metalloid Tolerance Genes and Mechanisms

### 3.1. Arsenic

The *arsABC* operon is a set of genes that give tolerance to arsenic oxyanion in both Gram-negative and Gram-positive bacteria such as *Staphylococcus*, *Bacillus*, *Rhodococcus*, and *E. coli* [196,211,212,213]. This operon is located in plasmids, as well as chromosomes, and encodes an arsenate reductase *asrC*, an enzyme that transforms AsO_4_^3−^ into the AsO_3_^3−^ form. Other protein encoded by this operon in *Rhodococcus* includes *asrB* (As (III) specific efflux integral membrane protein), *asrA* (Anion translocating ATPase protein, *asrA* + *asrB* constitute an efflux pump), *asrD* (Regulatory protein, a chaperon protein), and *asrR* (Regulatory protein, repressor) [196]. Entrance to the cell has been discussed [214]. GlpF is an aquaglyceroporin transporter specific for organic solutes such as urea and glycerol and is used to transport AsO_3_^3−^ across the membrane. Due to the structural similarity to phosphate, AsO_4_^3−^ uptake to the cell occurs through phosphate transporters, including *pit* (phosphate inorganic transport) and *pst* (phosphate-specific transport). The *pit* is the primary transporter protein that gains energy from the proton motive force. In contrast, *pst* is an ABC-type periplasmic transporter that facilitates the uptake of AsO_4_^3−^ less efficiently [215]. Down-regulation or mutations in these transporters can give rise to As oxyanion resistance. AsO_4_^3−^ can be an electron acceptor in dissimilatory AsO_4_^3−^ reduction and energy generated by coupling reduction of AsO_4_^3−^ with inorganic and organic substances’ oxidation. Arr, a heterodimer membrane-bound enzyme, catalyzes this process in *Desulfurispirillum indicum* strain S5. Arr enzymes constitute two catalytic subunits including ArrA (Molybdopterin center + [4Fe-4S] cluster) and ArrB ([Fe-4S] center + 3 or 4 [4Fe-4S] clusters). In the presence of oxygen and NO_3_^−^ as electron acceptors in the respiration process, the expression of Arr genes is repressed [215,216]. ArrA and ArrB also are found in both the anaerobic Gram-positive bacterium, *Chrysiogenes arsenatis*, and the Gram-negative bacterium *Shewanella* sp. Strain ANA-3 [217].

The expression of three genes, including ACR1, ACR2, and ACR3, provides the ability to tolerate arsenic oxyanions in *Saccharomyces cerevisiae*. ACR1 encoded a transcriptional regulator while ACR2 and ACR3 are arsenate reductase and plasma membrane AsO_3_^3−^ efflux pump, respectively. Overexpression of cytoplasmic thiols and pumping As (III) out of the cell followed by reducing As (V) is another resistance mechanism observed in *S. cerevisiae*. In addition to this mechanism, fungi can sequestrate arsenic oxyanions in vacuole intracellularly. Bacteria and fungi methylate arsenic oxyanions and transform them into volatile species to decrease their adverse effects. In a high concentration of AsO_4_^3−^, the level of GiPT transcript elevated, and a few hours later, GiArsA (putative AsO_3_^3−^ efflux) level increased. GiPT acts as a high-affinity phosphate transporter that uptake AsO_4_^3−^ into the cell. Mechanisms of AsO_4_^3−^ reduction in AMF is still unclear. It was suggested that an arsenate reductase might exist and transform AsO_4_^3−^ into AsO_3_^3−^. Then, AsO_3_^3−^ expels from the cell by GiArsA and GvArsA as AsO_3_^3−^ efflux pumps [218]. As a result of toxic metal accumulation, oxidative stress increases. Microorganisms have evolved some strategies to conquer the adverse consequences of oxidative stresses. In *Aspergillus niger*, the concentration of cytoplasmic thiols, malondialdehyde, proline, and anti-oxidative stress enzymes such as catalase, superoxide dismutase, and succinate dehydrogenase elevated as a result of an increase in AsO_3_^3−^ up to a certain level. In this study performed by Mukherjee et al. (2010), *A. niger* was exposed to different AsO_3_^3−^ doses ranging from 25 to 100 mg/L. The results showed that growth rate and thiol content were maximum at 25 mg/L and 100 mg/L AsO_3_^3−^, respectively. The most level of AsO_3_^3−^ absorption and intracellular AsO_3_^3−^ was observed at 75 mg/L.

Different efflux proteins such as ArsB, Acr3, ArsP, ArsJ, and MSF1 were evolved in arsenic-resistant bacteria for arsenic resistance. Some metalloids such as AsO_3_^3−^-, AsO_4_^3−^-, Sb(OH)_3_-, and methyl AsO_3_^3−^-induced the expression of *arsK*. *arsK* is a novel efflux system for metalloid transportation and is regulated by ArsR2. *arsK* and *arsR2* are integrated into one operon and transcribed under one promotor’s regulation [219]. Different mechanisms in the entrance and transportation of arsenic in *E. coli* and *S. cerevisiae* is summarized in Table 2. It is suggested that this mechanism also is used for antimony uptake [142].

Major intrinsic proteins (MIPs) are one of the transporter systems that participated in metalloid homeostasis in microorganisms. MIPs can expel them from the cell at a high level of metalloids and assimilate them when microorganisms need a beneficial metalloid. In *S. cerevisiae*, As (III) is conjugated to GSH, and Ycf1p has transported them by vacuolization. In *S. cerevisiae*, As (III) conjugated to GSH, and Ycf1p transported them by vacuolization. The presence of As (III) in the medium initiates a MAP kinase signaling cascade that can be started with As (III) and leads to inhibition of As (III) influx by Fps1p closure. Some genes are involved in metalloid uptake, which was concluded in Table 3.

### 3.2. Selenium

Studies showed that microorganisms often used sulfate-thiosulfate permease and the SulP- type permease to transport selenium compounds. In addition to these proteins, other research revealed that membrane protein YedE might also be applied for selenium oxyanion transport [231].

Transport of SeO_3_^2−^ in *E. coli* occurred by at least three systems, including Sulphate transport complex ABC, sulfate permease, and one or more unknown transporters that even worked when the other two transporters were inactivated. *cysAWTP* operon encodes sulfate transport complex ABC and consists of CysT and CysP. A polyol transporter found in *Rhodobacter sphaeroides*, is responsible for transferring SeO_3_^2−^ into cells [232,233].

Many genes were found that are involved indirectly in SeO_4_^2−^ reduction, such as *fnr*, *tatC*, and *menD*. These genes have central roles in expressing anaerobic genes, the exportation of folded proteins across the inner membrane, and the menaquinone biosynthesis pathway, respectively. Theisen and Yee (2014) found a functional selenate reductase gene, namely *ynfE*, in *Citrobacter fereundii* that encodes a molybdenum-binding Tat-secreted protein. An FNR binding site was found upstream of the *ynfE* gene, suggesting that SeO_4_^2−^ reduction in this bacterium occurred in anaerobic conditions. *ynfE* gene is located in the *ynfEGHdmsD* operon [234], and a high structural identity is observed between the selenate reductase of this bacterium and selenate reductases found in *Gammaproteobacteria* [235]. FNR is a general regulator system for fumarate and nitrate reductases and well regulates all CISMS (complex iron sulfur-molybdo enzymes). The menaquinone is the electron carrier to CISMS and TatC is a protein translocase that accepts CISM enzymes to help in assembly at the cytoplasmic membrane. Thus, both FNR and TatC are required for the YnfE protein to mature but have nothing specifically to do with SeO_3_^2−^ metabolism, and SeO_3_^2−^ can be processed by CISM-type enzymes including even NarGHI in *E. coli* [236].

In anaerobic conditions, the *fnr* gene is responsible for regulating selenate reductase activity in *Enterobacter cloacae* SLD1a-1. Recombinant *Escherichia coli* S17-1 containing the *fnr* gene from *E. cloacae* can reduce SeO_4_^2−^ to Se (0) with similar rates to those for *E. cloacae*. Even SeNPs biosynthesized by this bacterium have the same size and shape as SeNPs produced by *E. cloacae*. Lack of SeO_4_^2−^ reduction activity and SeNPs biosynthesis are observed in *E. cloacae* mutants in the *fnr* gene [237].

Another operon involved in SeO_4_^2−^ reduction is the *srdBCA* operon, which encodes a respiratory selenate reductase in Gram-positive bacteria such as *Bacillus selenatarsenatis*. Nitrate and nitrite reductase, protein, or peptide-containing thiol groups, chaperones, and some elongation factors also may be involved in selenium oxyanion transformation to Se (0) [238].

The SerABC complex is a soluble periplasmic enzyme in *Thauera selenatis*, consisting of three subunits, including SerA, SerB, and SerC, and a b-type cytochrome. In contrast, this complex in *E. cloaca* is a membrane-bound enzyme, and molybdenum, iron, and sulfur are the prosthetic groups of enzymes. SerABC complex is a selenate reductase that can only reduce SeO_4_^2−^ and showed substrate specificity [239,240].

The *ygfKLMN* operon encodes another selenate reductase in *E. coli* K12. The deletion of the *ygfk* gene in this operon results in the loss of SeO_4_^2−^ reduction in this strain. Besides enzymes involved in SeO_4_^2−^ and SeO_3_^2−^ reduction, the mutation in genes involved in electron carrier biosynthesis, such as menaquinone, can also affect selenium oxyanion’s bioreduction activity. This is because most all of the selenate and selenite reductases are complex iron-sulfur molybdenum enzymes that interact with the quinone pool anaerobically where menaquinone dominates over quinone [241]. Menaquinone biosynthesis is performed by *menFDHBCE* gene cluster, *menA*, and *ubiE*, *menD*, *menC*, or *menE* gene mutants were unable in SeO_4_^2−^ reduction [242].

### 3.3. Boron

The transportation of some metalloids such as AsO_3_^3−^, Sb(OH)_3_, boron, and silicon across the membrane facilitates by MIPs in yeast. Furthermore, it was suggested that nodulin 26-like intrinsic protein (NIP) subfamily proteins facilitate these metalloids’ transportation. Borate is a carbonate analog and can be transported across the membrane by bicarbonate transporter-like BOR1 in plants. YNL275w is a homolog of BOR1 in *Saccharomyces cerevisiae* and also can be applied as an efflux boron transporter in the yeast cell membrane [118]. In contrast, AsO_4_^3−^ and Sb(OH)_6_^−^ are phosphate analogs, and transcription of them is facilitated by phosphate transporters [243].

### 3.4. Tellurium

Different enzymes have been found in various organisms to participate in TeO_3_^2−^ reduction [244] and consist of: nitrate reductases (Nar and Nap) [245], catalase, NADH dehydrogenase (type II), and dihydrolipoamide dehydrogenase (lpdA), alkyl hydroperoxide reductase (AhpF), thioredoxin reductase (TrxB), dihydrolipoamide dehydrogenase, glutathione reductase (GorA), NADH: flavorubredoxin reductase (NorW), mercuric reductase (MerA), and the putative oxidoreductase YkgC [246]. *The tehAB* genes involved in TeO_3_^2−^ resistance is found on the chromosome of *E. coli* where *tehB* is S-adenosylmethionine cofactor dependent and likely a telluromethylase [247,248]. In addition to the phosphate transporter, TeO_3_^2−^ can enter the cell via a monocarboxylate transporter in *R. capsulatus*, and acetate, lactate, and pyruvate can compete with TeO_3_^2−^ in this process [249]. The *ter* operon consists of seven genes, ZABCDEF, but little is understood about the mechanism, yet it seems to be associated with the *E. coli* pathogenicity island [250]. The *terD* gene seems to be able to give resistance on its own and is found in other species [250,251,252]. In another study, Maltman et al. (2017) found a new tellurite reductase with a 117 kDa molecular weight. In addition to TeO_3_^2−^, this membrane-associated enzyme can reduce TeO_4_^2−^ [253]. Other gene involved in TeO_3_^2−^ resistance includes *kilAtelAB* [254]. Many of these are found on large conjugative plasmids [255].

Some transporters and pathways in metalloid biotransformation in bacteria and fungi were summarized in Figure 2 and Figure 3, respectively.

## 4. Remediation under More Than One Metal (loid) Challenge

For bioremediation of contaminated areas with two or more metalloids, it is essential to know the microbial diversity and type of bacteria present in these regions. Indigenous microorganisms have different mechanisms for tolerating high levels of these contaminants, so their potential can be used for bioremediation. For example, the diversity of diazotrophic bacteria was investigated in two mines in China with high levels of arsenic and antimony. Due to low amounts of nitrogen, diazotrophic bacteria are important, as they are capable of both nitrogen fixation and metalloid tolerance in these regions. The most abundant diazotroph phyla include *Pseudomonadota* and *Cyanobacteria*, and *Sinorhizobium*, *Dechloromonas*, *Trichormus*, *Herbaspirillum*, *Desmonostoc*, and *Klebsiella* are prevalent genera in these areas [256].

Liu et al. (2017) investigated the simultaneous bioremediation of wastewater containing arsenic and antimony oxyanions by SRB. They used lactate and ethanol as carbon sources. Results showed that ethanol was a better electron donor and carbon source than lactate and it might be related to the lower amount of sulfide formation by ethanol in comparison with lactate consumption. In presence of both HAsO_4_^2−^ (As (V)) and Sb(OH)_6_^−^ (Sb (V)), the rate of As (V) removal was low and it improved remarkably by adding ferrous ions (Fe (II)). They suggested that sorption/co-precipitation by FeS is involved in increasing As (V) remediation by SRB [257].

An interesting response of microorganisms is observed in response to the simultaneous presence of contaminants such as TeO_3_^2−^ and SeO_3_^2−^ or TeO_3_^2−^ and mercury. In both cases, the presence of mercury or SeO_3_^2−^ with TeO_3_^2−^ increases the bacterial resistance to TeO_3_^2−^ in some microorganisms. The mechanism and cause of these reactions are still unknown [258,259]. In fact, it appears that SeO_3_^2−^ will actually protect against TeO_3_^2−^ challenge [260].

## 5. Metalloid Reduction through Biodegradation of Associated Organic Compounds

Microbial tolerance to metalloids has been investigated for over 40 years. Most of these investigations have concentrated on planktonic forms of single-species microorganisms [261,262,263,264,265,266], while microbial consortia and biofilms are the common forms of microorganisms present in the environment where “natural” exposure to metalloids occur [256,267,268,269,270,271,272,273]. A microbial biofilm comprises one or more species generally attached to a surface and in addition to the microbial cells, contains a considerable amount of extracellular matrix of exopolymeric substance (EPS; also referred to as exopolysaccharides). The matrix is a mixture of various biochemical polymers (nucleic acids, proteins, carbohydrates—EPS often also refers to exopolysaccharides) [274]. This matrix is of a defense nature but can also be used as a carbon source as well as a source of functional groups to bind ions and organic compounds. It acts as a barrier to the entry of some inhibitory compounds such as toxins and enables the exchange of electron and genetic information thorough its matrix [275]. Studies have shown that biofilms have a remarkable ability to remove various contaminants than planktonic species [276,277]. In addition to biofilm, microorganisms can usually form microbial consortia which referred to communities composed of more than two different species of microorganisms that can be used for different biotechnological applications such as bioremediation. These consortia can tolerate and remove significant amounts of pollutants rather than pure culture because of synergistic interactions between microbial species composing these consortia [278,279,280,281,282,283,284,285].

The idea of a biofilm community is illustrated by the ability of a consortium consisting of methanotrophic bacteria and archaea to reduce SeO_4_^2−^. The use of consortium resulted in a 1.8-fold increase in SeO_4_^2−^ reduction compared to the use of each of the bacterial and archaeal species separately [286]. This subject gained limited attention in the literature at the time. Precise underlying mechanisms are still unknown but now has attracted much more interest in recent years.

For designing optimized co-culture, a great extent of information is required about microbial community structure, metabolic requirements, interactions, and mechanisms. Consortia are composed of greater than two different microbial species that co-exist and interact together. These interactions between microorganisms are always negative, positive, or neutral and are categorized into six different forms: cooperation (commensalism and mutualistic relationships), prey and predator or parasite and its host, neutral interaction, competition, and amensalism. Cooperation is the best mode for consortia formation because of cross-feeding (syntropy) and metabolite exchange between these consortia [287,288].

In an exciting state of cooperation, one microorganism produces compounds and provides them to another microorganism, while another microorganism fulfills another microbe’s requirements by performing a similar action [289,290]. A microorganism may produce growth factors or enzyme-inducing compounds that cause more growth or more production rates in another microorganism. Sometimes microorganisms produce more products or increase the biomass of other microbes present in the consortium by removing or reducing stress factors such as toxins or inhibitors [291,292,293]. In certain cases, a competitive or antagonistic relationship between microorganisms leads to the production of beneficial products. Finally, the degradation of complex biodegradable compounds such as hydrocarbons by a microorganism in the consortium leads to smaller and easier intermediates for the consumption of other microbes and the formation of a metabolic network between the members of the consortium [288]. The summary of these mechanisms is illustrated in Figure 4.

Some microorganisms can couple reductions of metalloid oxyanions with hydrocarbon oxidation. In this regard, Luo et al. (2018) studied the bioreduction of SeO_4_^2−^ using denitrifying anaerobic methane-oxidizing biofilm. SeO_4_^2−^ was added to culture media with a concentration of 20–60 µmol/L in a membrane bioreactor (MBR) bioreactor with a reduced rate of 2.8–12.4 µmol/L/d. In the lack of methane, no SeO_4_^2−^ was reduced. Fluorescence in-situ hybridization (FISH) and 16S rRNA gene sequencing analysis were applied to detect biofilm microbial community structure. Results showed that Candidatus *Methanoperedens nitroreducens* and Candidatus *Methylomirabilis oxyfera* were dominant SeO_4_^2−^ reducers in the presence of methane as a carbon source [294].

In addition to SeO_4_^2−^, the reduction of SeO_3_^2−^ also can be coupled by the oxidation of methane. In this regard, Bai et al. (2019) used denitrifying methanotrophic microorganisms for bioreduction. The addition of NO_3_^−^ can inhibit the process. The authors proposed two hypothetical mechanisms for this process: methane oxidation by methanotrophs and the transfer of electrons to SeO_3_^2−^ by the same microorganism. The other involves a synergistic relationship between methanotrophs (electron transfer) and another microbe that can reduce SeO_3_^2−^ [295].

Liu et al. (2013) investigated the simultaneous removal of pyrene and arsenic by bacterial and fungal co-culture. This co-culture consists of two bacterial strains, namely, *Bacillus* and *Sphingomonas*, and *Fusarium* as a fungus. They made five different culture conditions; culturing isolates independently, bacterial culture, and a combination of three strains with arsenic (20 µM) as an energy source and pyrene (100 mg/L) as a carbon source. Results showed that the highest amount of pyrene degradation and arsenic methylation occurred in the presence of a mix-culture of all three strains because of the synergistic effect. The bioremediation of pollutants by co-culture was also decreased to nine days compared to 63 days As volatilization time by bacterial strains (only 13.9% arsenic removal) [296].

Due to diverse reductase enzymes, strains of the genus *Pseudomonas* strains are competent bacteria in the bioremediation of contaminants. In a study conducted by Feng et al. (2014), these strains’ ability to remove AsO_4_^3−^ and poly aromatic hydrocarbons (PAH) was investigated individually and concurrently. These strains are capable of complete phenanthrene (of 60 mg/L) and half pyrene (20 mg/L) elimination in 60 h and reduction of 1.5 mM AsO_4_^3−^ in 48 h. The arsenic reduction rate will be significantly decreased, while PAH is exploited as the sole source of carbon and energy. The addition of lactate as an electron source increases the ability to degrade simultaneous pyrene and AsO_4_^3−^ reduction in *Pseudomonas* strains [297].

Despite the biodegradation of a broad spectrum of organic contaminants, Methanotrophs can transform inorganic pollutants such as SeO_3_^2−^ into safe elemental Se. In this regard, Eswayah et al. (2017) studied the ability of pure culture of *Methylococcus capsulatus* and *Methylosinus trichosporium* in simultaneous methane oxidation and SeO_3_^2−^ remediation. GC-MS analysis showed that these bacteria could volatize SeO_3_^2−^ in addition to producing elemental Se. Localization of elemental Se showed that this process occurred on the bacterial cell wall. Volatile Se forms produced by *Mc. capsulatus* include DMSe (di methyl selenide), DMDSe (di methyl di selenide), DMSeS (dimethyl selenenyl sulfide), Methyl selenol, and Methylseleno acetate while *Ms. trichosporium* can only volatize SeO_3_^2−^ into DMDSe, and DMSeS [298]. Although the pure culture of these two bacteria could not transform SeO_4_^2−^ into its non-toxic form, another study by Lai et al. (2016) showed the ability of cultured mixtures of methanotrophs to remove SeO_4_^2−^. In this study, the complete bioremediation of SeO_4_^2−^ was achieved using methane as a carbon source and electron donor in a membrane bioreactor (MBfR). Additionally, the effect of NO_3_^−^ was investigated on the SeO_4_^2−^ bioreduction process. Results showed that NO_3_^−^ could interfere with SeO_4_^2−^ bioreduction and inhibit this process slightly. The simultaneous reduction of NO_3_^−^ and SeO_4_^2−^ occurred even when the amount of methane was low. Authors suggested two different mechanisms for this process: i. some methanotroph microorganisms such as *Methylomonas* can cause couple reduction in SeO_4_^2−^ with the oxidation of methane. ii. The synergistic relationship between methanotrophs and SeO_4_^2−^ reducing microorganisms in this way that some methanotroph microorganisms can oxidize methane and provide electrons for other microorganisms that can reduce senate into elemental selenium (Se0) [299].

Since AsO_4_^3−^ is less toxic than AsO_3_^3−^, AsO_3_^3−^-oxidizing microorganisms can also be used for bioremediation purposes. In a study conducted by Tang et al. (2013), the ability of a heterotrophic bacterial consortium in simultaneous phenanthrene biodegradation and AsO_3_^3−^ oxidation was investigated. This consortium can remediate 71.4% of phenanthrene (200 mg/L) and oxidize 96.2% of AsO_3_^3−^ (60 mg/L) after 48 h incubation simultaneously. Dominant genera of this consortium include *Pseudomonas*, *Pusillimonas*, *Alcaligenes*, and *Achromobacter*. The shift in this consortium’s community structure in the presence of contaminates is listed in Table 4 [300].

Mix-culture of sulfate-reducing bacteria (SRB) was isolated from an Sb mine in Guangxi, China, and applied to eradicate antimony and arsenic co-contaminants from wastewater. In this study, Liu et al. (2018) used ethanol and lactate as carbon sources and electron donors. The amount of arsenic and antimony removal increased in the presence of ethanol rather than lactate, indicating that ethanol is a better carbon source than lactate. After 12 days, this mixed culture can reduce 97.8 and 26.4% of 5 mg/L of both antimony and arsenic in the presence of ethanol, respectively. The addition of Fe (II) can elevate these amounts to 99.4% (Antimony) and 98.2% (arsenic) through co-precipitation [257].

Zhang et al. (2016) applied a mixed SRB culture for antimony bioremediation of a stream contaminated by coalmine effluents. This mixed culture could remediate 93% of Sb (V) (5 mg/L) over 11 days of incubation. Due to bioreduction, insoluble antimony sulfide (Sb_2_S_3_) was produced and detected microscopically. Further studies showed that microbial biosorption had an insignificant role in the antimony bioreduction of this SRB mix culture [301].

By producing EPS, many microorganisms can withstand environmental stress and the invasion of bacteriophages, and the effects of certain toxins, creating anaerobic chambers and separating aerobic and anaerobic microenvironments. This is especially important in bioremediation, where some microorganisms grow under anaerobic conditions and use oxyanions as the last electron acceptor in respiratory reactions. Besides, EPS can be used as a carbon source by microorganisms. It can also act as a barrier against the loss of valuable compounds around microorganisms. Since metalloid bioremediation is based on redox reactions, the presence of structures such as nanowires that form between bacteria and archaea and are involved in electron transfer should also be investigated in studies [302,303,304].

## 6. Conclusions

Despite the considerable research to date that has been completed on the bioremediation of metalloids so far, we still have poor knowledge about the underlying mechanisms and requirements of the microorganisms involved in this process. Proteomics and metabolomics studies can be perfect methods for illustrating the involved mechanisms of bioremediation. Furthermore, some software can be designed for predicting the suitable amount of nutrients, enzyme cofactors, electron shuttles, type of microorganisms, and physiochemical factors that are required for efficient ex-situ bioremediation of polluted sites. The relationship between sulfur and nitrogen cycles with the detoxification of metalloids and the role of microorganisms involved in these two cycles in the bioremediation of contaminated areas are fascinating and more investigations are needed in this field of study. Moreover, proteomic and transcriptomic studies can help reveal the synergistic relationships of microorganisms that reduce metalloids and decrease their toxic effects.

## Figures and Tables

**Figure 1 microorganisms-11-00891-f001:**
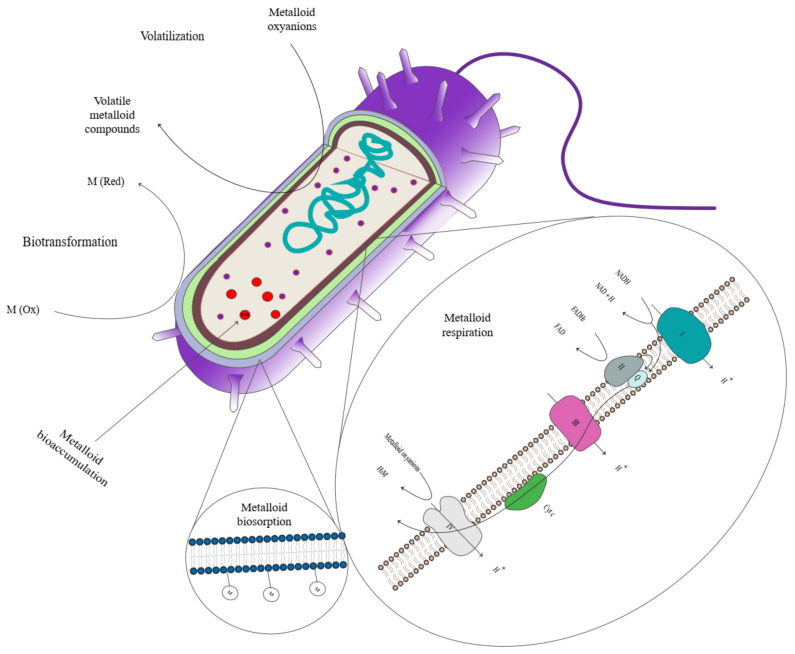
General mechanisms of metalloid tolerance in microorganisms.

**Figure 2 microorganisms-11-00891-f002:**
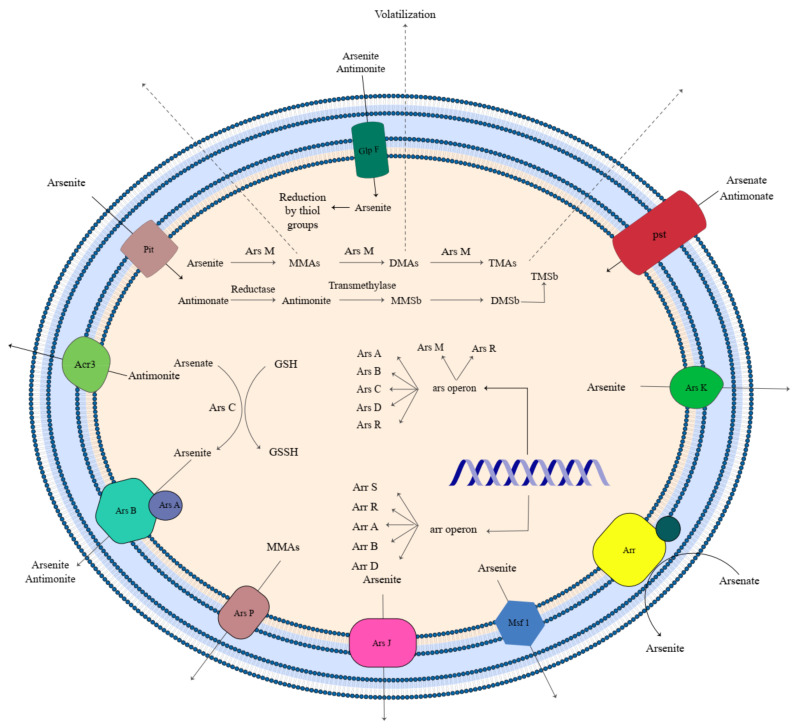
Arsenic and antimony transporters and remediation mechanisms in bacteria (Although this microorganism is shown as a Gram-negative bacterium, transporters and mechanisms belong to different species of Gram-positive and Gram-negative bacteria).

**Figure 3 microorganisms-11-00891-f003:**
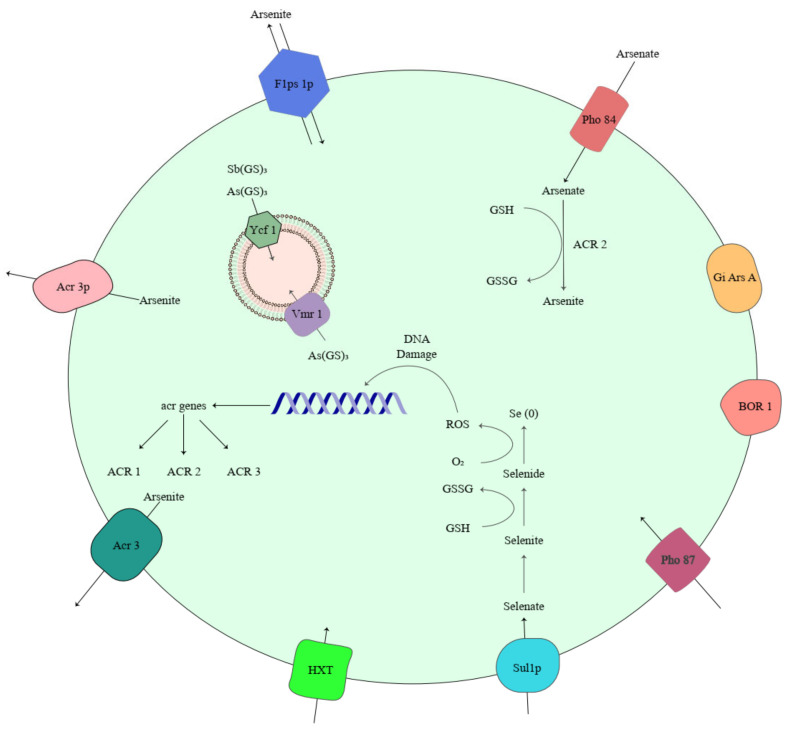
Metalloid transporter and protein involved in metalloid uptake and detoxification in fungi.

**Figure 4 microorganisms-11-00891-f004:**
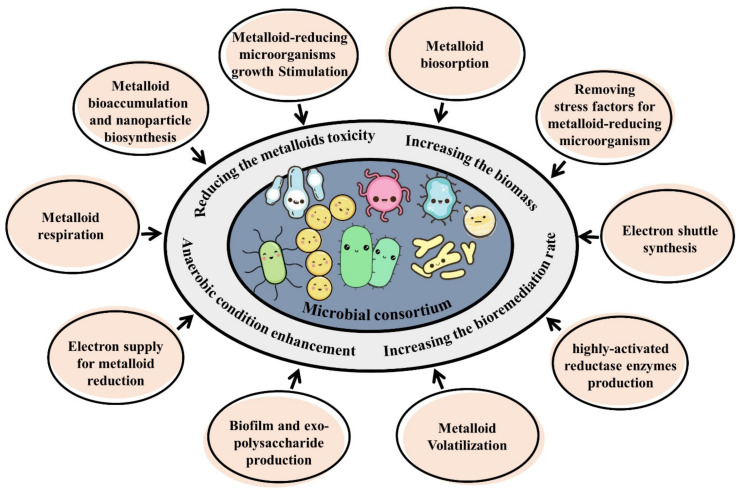
Some mechanisms proposed to attenuate the toxicity of metalloids in the environment.

**Table 1 microorganisms-11-00891-t001:** The ecology of arsenic and selenium-resistant bacteria in different contaminated sites.

Location	Sample Site	Phylum/Class	Genus/Species	Ref.
Louisiana, USA	Lake Pontchartrain estuary	*Gammaproteobacteria*	*Aeromonas*, *Rheinheimera*, *Pseudomonas*	[203]
		*Betaproteobacteria*	*Rhodoferax*, *Variovorax*, *Rubrivorax*, *Janthinobacterium*, *Zoogloea*	
		*Alphaproteobacteria*	*Brevundimonas*, *Agrobacterium*, *Aquaspirillum*, *Sphingomonas*, *Porphyrobacter*	
		*Bacillota*	*Bacillus*	
		*Actinomycetota*	*Curtobacterium*	
		*Bacteroidota*	*Flavobacterium*	
Taiwan	Chianan plain, soil, and groundwaters	*Alphaproteobacteria*	*Sphingomonas*, *Microvirga*, *Phenylobacterium*, *Ochrobactrum*, *Sphingosinicella*	[204]
		*Gammaproteobacteria*	*Pseudomonas*, *Rheinheimera*, *Steroidobacter*, *Marinobacter*, *Azotobacter*, *Lysobacter*, *Acinetobacter*	
		*Betaproteobacteria*	*Janthinobacterium*, *Azonexus*, *Azohydromonas*, *Herbaspirillum*, *Massilia*	
		*Deltaproteobacteria*	*Syntrophobacter*	
		*Bacillota*	*Bacillus*, *Clostridium*, *Tumebacillus*, *Vulcanibacillus*, *Ammoniphilus*, *Shimazuella*, *Paenibacillus*, *Falsibacillus*, *Anaerobacter*,	
		*Actinomycetota*	*Marmoricola*, *Nocardioides*, *Pimelobacter*, *Arthrobacter*,	
		*Bacteroidota*	*Flavisolibacter*, *Pontibacter*, *Parasegetibacter*, *Ohtaekwangia*, *Pedobacter*,	
		*Chloroflexota*	*Sphaerobacter*	
		*Nitrospiria*	*Nitrospira*,	
		*Gemmatimonadota*	*Gemmatimonas*,	
India	Groundwater of Asam	*Alphaproteobacteria*	*Brevundimonas*, *Bosea*, *Caulobacter*, *Rhizobium*	[205]
		*Betaproteobacteria*	*Acidovorax*, *Undibacterium*, *Herbaspirillum*, *Ralstonia*	
		*Gammaproteobacteria*	*Acinetobacter*, *Pseudomonas*	
		*Actinomycetota*	*Rhodococcus*	
		*Bacillota*	*Staphylococcus*, *Bacillus*	
India	Nawanshahr in Punjab, Maharashtra, and Andhra Pradesh	*Betaproteobacteria*	*Achromobacter xylosoxidans*, *Delftia tsuruhatensis*, *Delftia tsuruhatensis*	[191]
		*Gammaproteobacteria*	*Pseudomonas* sp., *Enterobacter cowanii*	
		*Bacilli*	*Bacillus fusiformis*, *Bacillus* sp., *Bacillus sphaericus*	

**Table 2 microorganisms-11-00891-t002:** Comparison of arsenic transportation in bacteria and yeast.

Function	*E. coli*	*S. cerevisiae*	Ref.
As (V) entrance to cells	Phosphate transporters (PiT and Pst)	Phosphate transporters (Pho84p)	[220,221]
As resistance operon	*Ars*	Gene cluster similar to *ars*	[213,222]
As (III) extrusion transporter (Active transportation)	ATP-fueled (ArsA and ArsB)	H^+^-coupled (Acr3p) (Proton gradient generated by plasma membrane proton ATPase Pma1p)	[223,224]
As (III) efflux (Passive transportation)	MIP (GlpF)	MIP (Fps1p)	[224,225,226]
As (V) to As (III) reduction	ArsC	Acr2p	[227,228]

**Table 3 microorganisms-11-00891-t003:** Genes involved in metalloid uptake in *S. cerevisiae* [229,230].

Gene/Protein	Deletion	Overexpression	Ref.
BOR1	More susceptible rather than wild type to the toxic level of boron	Increases resistance to boric acid and decreases efflux rate	[229,230]
FPS1	Reduction of boron accumulation	No effect on the accumulation	
DUR3	Reduction of boron accumulation	No effect on the accumulation	
fps1-Δ1 allele lacking N-terminal extension	-	hypertensive to As (III) and Sb (III) and accumulated more As (III)	
FPS1	Increased resistance to As (III) and Sb (III)	-	
HOG1	Increased sensitivity to As (III) and Sb (III) due to upregulation of metalloid intake	-	

**Table 4 microorganisms-11-00891-t004:** Dominant taxa in the presence of contaminants.

Contaminants	Dominant Groups in the Presence of Contaminants	Ref.
Phenanthrene	*Pseudomonas* and *Achromobacter* spp.	(Tang et al., 2013) [300]
AsO_3_^3−^	*Pseudomonas* and *Alcaligenes* spp.	
Phenanthrene + **AsO_3_^3−^**	*Gammaproteobacteria* (*Pseudomonas*) and *Bettaproteobacteria* (*Achromobacter* and *Alcaligenes*)	

## Data Availability

Data is contained within the article.

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
