# Peer review of "Use of Microbial Consortia in Bioremediation of Metalloid Polluted Environments"

_microorganisms, 2023, doi:10.3390/microorganisms11040891_

Round 1

Reviewer 1 Report

Authors

The manuscript reviews recent findings in the microbial bioremediation of metalloids, including As, B, Ge, Sb, Se, Si, and Te. The manuscript describes the environmental impacts related to metalloid pollution, principal microorganisms identified with high potential for metalloid bioremediation as well as the genes and enzymes implicated in the process.

The manuscript includes valuable information related to metalloid bioremediation and has relevance to scientific readers in the field of microbial metal/metalloid bioremediation. The manuscript could be considered for acceptance after addressing the following commentaries.

Main Commentaries:

Abstract/Introduction: In the abstract and introduction, describe de gaps in the scientific reports about metalloid removal by microorganisms, and that were addressed by the present manuscript.

Conclusions: According to the review carried out, what could be the mechanisms and requirements for the educational bioremediation of metalloids?

Additional commentaries:

Lines 3-6, correct the format in the Author´s information

Line 14, choose a synonym for “creating”, that better suit to the main idea of the threats to human health

Line 40, define the use of hypertext for “th” or not, and homogenize in the whole manuscript.

Line 68, add a space between “AsO33-is”

Line 77: Define the acronym PNS (Peripheral Nervous System Disorders) first time used.

Line 89: The essential term is repeated twice on the same line; it is suggested to change it to another word

Lines 128-129, the texts in highlighted in gray

Line 142, could be “nano” instead “Nano”

Lines 144-145, the text is highlighted in gray

Lines 160-163, the text is highlighted in gray

Lines 164-189, the text is highlighted in gray

Line 178: Add space between paragraphs

Line 231, 376, 377, 378, 573, 608, 609, 617: Homogenize the units mg/L, mg/l or mg.l-1 in the document, review carefully whole manuscript

Line 233, define the acronym “FAME”

Line 280, add space between paragraphs

Line 293, add space between paragraphs

Line 302, add space between paragraphs

Line 310, add space between paragraphs

Line 315, add space between paragraphs

Line 321, 576, 608, 616, 618, remove the space between the number and the percentage (0.4 %, 13.9 %, 71.4, 96.2 %, 26.4 %), review carefully whole manuscript

Line 325, add space between paragraphs

Line 339, could be “…plasmids, as well as chromosome,”

Line 342, remove extra parenthesis

Line 358, add space between paragraphs

Line 379, add space between paragraphs

Line 403, “hypersensitive” could be “Hypertensive”

Lines 406 and 408, “increased” could be “Increased”

Line 421, include the year in the reference (Theisen and Yee ….)

Line 432, add space between paragraphs

Line 442, write the number 3 in letters

Line 446, add space between paragraphs

Line 454, remove the point and put a comma after ubiE

Line 539, add space between paragraphs

Line 568, add space between paragraphs

Line 569, include the year in the reference Liu et al.

Line 575, write the number 9 in letters

Line 584, add space between paragraphs

Line 586. include the year in the reference (Eswayah et al., ….)

Line 594, include the year in the reference (Lai et al., ….)

Line 600, eliminate extra comma

Line 604, remove extra parenthesis

Line 611, add space between paragraphs

Line 613, include the year in the reference Liu et al.

Line 619, include the year in the reference Zhang et al.

Line 623, add space between paragraphs

Author Response

point-by-point response

Reviewer 1

The manuscript reviews recent findings in the microbial bioremediation of metalloids, including As, B, Ge, Sb, Se, Si, and Te. The manuscript describes the environmental impacts related to metalloid pollution, principal microorganisms identified with high potential for metalloid bioremediation as well as the genes and enzymes implicated in the process.

The manuscript includes valuable information related to metalloid bioremediation and has relevance to scientific readers in the field of microbial metal/metalloid bioremediation. The manuscript could be considered for acceptance after addressing the following commentaries.

Thanks to the honorable reviewer of the article.

(The revised/changed sections in the manuscript are highlighted with turquoise.)

Main Commentaries:

Abstract/Introduction: In the abstract and introduction, describe the gaps in the scientific reports about metalloid removal by microorganisms, and that were addressed by the present manuscript.

Thanks to your valuable comment, some sentences were added to abstract and introduction to cover the gaps in scientific reports that addressed in this study (Lines 20-22, 193-197).

Conclusions: According to the review carried out, what could be the mechanisms and requirements for the educational bioremediation of metalloids?

The information was added to the conclusion (Lines 643-647).

Additional commentaries:

Lines 3-6, correct the format in the Author´s information

The format of Author’s information was corrected (Lines 3-6).

Line 14, choose a synonym for “creating”, that better suit to the main idea of the threats to human health

The word of “creating” were replaced with “causing” (Line 14).

Line 40, define the use of hypertext for “th” or not, and homogenize in the whole manuscript.

The hypertext was used for all “th” in manuscript (Lines 42, 62, 82, 122, and 139).

Line 68, add a space between “AsO33-is”

The space was added (Line 70).

Line 77: Define the acronym PNS (Peripheral Nervous System Disorders) first time used.

The definition of the acronym PNS (Peripheral Nervous System Disorders) was added (Line 79).

Line 89: The essential term is repeated twice on the same line; it is suggested to change it to another word

The word of “essential” was replaced with “pivotal” in this sentence (Line 91).

Lines 128-129, the texts in highlighted in gray

Thanks for your correction, the highlight was removed (Line 130-131).

Line 142, could be “nano” instead “Nano”

The lowercase was used instead (Line 144).

Lines 144-145, the text is highlighted in gray

Lines 160-163, the text is highlighted in gray

Lines 164-189, the text is highlighted in gray

Thanks for your correction, the highlights were removed in all sentences (Lines 145-146, 166-169, 170-200).

Line 178: Add space between paragraphs

The space between all paragraphs were corrected (Line 183).

Line 231, 376, 377, 378, 573, 608, 609, 617: Homogenize the units mg/L, mg/l or mg.l-1 in the document, review carefully whole manuscript

All units were unified as mg/L (Lines 242, 386, 387, 388, 579, 587, 588, 615, 616, and 627).

Line 233, define the acronym “FAME”

This acronym was defined as “Fatty Acid Methyl Ester” in manuscript (Line 244).

Line 280, add space between paragraphs

Line 293, add space between paragraphs

Line 302, add space between paragraphs

Line 310, add space between paragraphs

Line 315, add space between paragraphs

The space between all paragraphs were corrected (Lines 290, 303, 312, 320, and 325).

Line 321, 576, 608, 616, 618, remove the space between the number and the percentage (0.4 %, 13.9 %, 71.4, 96.2 %, 26.4 %), review carefully whole manuscript

The space between the number and the percentage in whole manuscript was removed (Lines (331, 615, 623, and 625).

Line 325, add space between paragraphs

The space between all paragraphs were corrected (Line 335).

Line 339, could be “…plasmids, as well as chromosome,”

The right substitution was replaced by suggestion of reviewer (Line 349).

Line 342, remove extra parenthesis

Extra parenthesis was removed (Line 352-353).

Line 358, add space between paragraphs

Line 379, add space between paragraphs

The space between all paragraphs were corrected (Line 368, and 388).

Line 403, “hypersensitive” could be “Hypertensive”

The word of “hypertensive” was replaced with “hypersensitive” (Line 412).

Lines 406 and 408, “increased” could be “Increased”

The word of “Increased” was replaced with “increased” in the Table (Lines 415, and 417).

Line 421, include the year in the reference (Theisen and Yee ….)

The reference was corrected (Line 430).

Line 432, add space between paragraphs

The space between all paragraphs were corrected (Line 441).

Line 442, write the number 3 in letters

Number 3 was replaced with three in the manuscript (Line 451).

Line 446, add space between paragraphs

The space between all paragraphs were corrected (Line 455).

Line 454, remove the point and put a comma after ubiE

A comma was added after ubiE (Line 462).

Line 539, add space between paragraphs

Line 568, add space between paragraphs

The space between all paragraphs were corrected (Lines 546, and 574).

Line 569, include the year in the reference Liu et al.

The year was added to reference (Line 575).

Line 575, write the number 9 in letters

Number 9 was written in letters (nine) (Line 582).

Line 584, add space between paragraphs

The space between all paragraphs were corrected (Line 591).

Line 586. include the year in the reference (Eswayah et al., ….)

The year was added to reference (Line 594).

Line 594, include the year in the reference (Lai et al., ….)

The year was added to reference (Line 601).

Line 600, eliminate extra comma

Extra comma was removed (Line 607).

Line 604, remove extra parenthesis

Extra parenthesis was eliminated (Line 611).

Line 611, add space between paragraphs

The space between all paragraphs were corrected (Line 618).

Line 613, include the year in the reference Liu et al.

The year was added to reference (Line 621).

Line 619, include the year in the reference Zhang et al.

The year was added to reference (Line 626).

Line 623, add space between paragraphs

The space between all paragraphs were corrected (Line 630).

Reviewer 2 Report

Title:

Use of microbial consortia in bioremediation of metalloid polluted environments.

Recommendation:

    Minor revision.

Comments:

This manuscript reviewed the articles which used microbial consortia in bioremediation of metalloid polluted environments. The subject is relevant and consistent with the aims and scopes of the journal. In my opinion, the manuscript is interesting and organizes many research results and concepts related to microbial consortia and metalloid. Some comments and suggestions are offered below with the intent to assist the author in improving the manuscript.

1.     On the paragraph number, there are two "1.2.2". Please revise this part.

2.     Because in the manuscript, there are many narratives mentioning "tolerance genes and mechanisms". Therefore, it is suggested that some descriptions can also be added in the part of abstract and keywords.

3.     There are still many typo or formatting errors in the manuscript, such as "Bai et al. (2019)" at line 564. Therefore, please revise them carefully.

4.     There are still many formatting errors in the references section, such as the year not being bolded. So please revise them carefully.

Author Response

point-by-point response

Reviewer 2

This manuscript reviewed the articles which used microbial consortia in bioremediation of metalloid polluted environments. The subject is relevant and consistent with the aims and scopes of the journal. In my opinion, the manuscript is interesting and organizes many research results and concepts related to microbial consortia and metalloid. Some comments and suggestions are offered below with the intent to assist the author in improving the manuscript.

Thanks to the honorable reviewer of the article. (The revised/changed sections in the manuscript are highlighted with bright green.) 

  1. On the paragraph number, there are two "1.2.2". Please revise this part.

Thanks for your correction, the paragraph number was edited (Lines 81, 101, 120, and 137).  

  1. Because in the manuscript, there are many narratives mentioning "tolerance genes and mechanisms". Therefore, it is suggested that some descriptions can also be added in the part of abstract and keywords.

Thanks for your suggestion, the information was added to abstract and keywords (Lines 19, and 27).

  1. There are still many typo or formatting errors in the manuscript, such as "Bai et al. (2019)" at line 564. Therefore, please revise them carefully.

Formatting errors and typos were corrected. Also, the year was added to some references (Line 570).

  1. There are still many formatting errors in the references section, such as the year not being bolded. So please revise them carefully.

Thanks for your suggestion, all years in references have been bolded (Lines 662-1446).
